# Trial Of Neurostimulation In Conversion Symptoms (TONICS): a feasibility randomised controlled trial of transcranial magnetic stimulation for functional limb weakness

Susannah Pick [1], John Hodsoll [1], Biba Stanton [1,2], Amy Eskander [2], Ioannis Stavropoulos [3], Kiran Samra [2], Julia Bottini,[2] Hena Ahmad,[2] Anthony S David [4], Alistair Purves [3], Timothy R Nicholson [1]

► Prepublication history and supplemental material for this paper are available online. To view these files, please visit the journal online (http://dx.doi.org/10.1136/bmjopen-2020-037198).

For numbered affiliations see end of article.

**Correspondence to**
Dr Timothy R Nicholson;
timothy.nicholson@kcl.ac.uk

## ABSTRACT

**Objectives** Transcranial magnetic stimulation (TMS) has been used therapeutically for functional (conversion) motor symptoms but there is limited evidence for its efficacy and the optimal protocol. We examined the feasibility of a novel randomised controlled trial (RCT) protocol of TMS to treat functional limb weakness.

**Design** A double-blind (patient, outcome assessor) two parallel-arm, controlled RCT.

**Setting** Specialist neurology and neuropsychiatry services at a large National Health Service Foundation Trust in London, UK.

**Participants** Patients with a diagnosis of functional limb weakness (Diagnostic and Statistical Manual of Mental Disorders - Fifth Edition). Exclusion criteria included comorbid neurological or major psychiatric disorder, contraindications to TMS or previous TMS treatment.

**Interventions** Patients were randomised to receive either active (single-pulse TMS to primary motor cortex (M1) above resting motor threshold) or inactive treatment (single-pulse TMS to M1 below resting motor threshold). Both groups received two TMS sessions, 4 weeks apart.

**Outcome measures** We assessed recruitment, randomisation and retention rates. The primary outcome was patient-rated symptom change (Clinical Global Impression–Improvement scale, CGI-I). Secondary outcomes included clinician-rated symptom change, psychosocial functioning and disability. Outcomes were assessed at baseline, both TMS visits and at 3-month follow-up.

**Results** Twenty-two patients were recruited and 21 (96%) were successfully randomised (active=10; inactive=11). Nineteen (91%) patients were included at follow-up (active=9; inactive=10). Completion rates for most outcomes were good (80%–100%). Most patients were satisfied/very satisfied with the trial in both groups, although ratings were higher in the inactive arm (active=60%, inactive=92%). Adverse events were not more common for the active treatment. Treatment effect sizes for patient-rated CGI-I scores were small-moderate (Cliff's delta=−0.1–0.3, CIs−0.79 to 0.28), reflecting a more positive outcome for the active treatment (67% and

### Strengths and limitations of this study

► The study examined the feasibility of a novel, controlled transcranial magnetic stimulation (TMS) protocol for treating functional limb weakness.
► The TMS protocol has potential to inform the minimal dose required and mechanism of action for positive outcomes in this population.
► Both patients and outcome assessors were blinded to treatment allocation, but it was not possible to blind the delivery of the treatment.
► As this was a feasibility study with a small sample size, randomisation might not have adequately balanced group differences across the treatment arms.

44% of active arm-rated symptoms as 'much improved' at session 2 and follow-up, respectively, vs 20% inactive group). Effect sizes for secondary outcomes were variable.

**Conclusions** Our protocol is feasible. The findings suggest that supramotor threshold TMS of M1 is safe, acceptable and potentially beneficial as a treatment for functional limb weakness. A larger RCT is warranted.

**Trial registration number** ISRCTN51225587.

## BACKGROUND

Functional neurological disorder (FND) is defined by neurological symptoms that are incompatible with other medical/neurological diagnoses.[1] FND can resemble any neurological disorder, with seizures, motor (eg, limb weakness, tremor, dystonia, myoclonus) and sensory (visual, auditory, somatosensory) symptoms predominating. Quality of life and prognosis are often poor.[2–4] Despite recent developments in detection and diagnosis of the disorder,[5] there is still a marked paucity of evidence-based, accessible treatment options. There is emerging evidence for the efficacy of some treatment modalities (eg,

specialist physiotherapy for motor symptoms or cognitive–behavioural therapy for seizures),[6–9] but availability is currently limited. The development of alternative treatment options that are safe, cost-effective, acceptable to patients and accessible is critical for improving outcomes in this population.

Transcranial magnetic stimulation (TMS) has been explored as a potential treatment option for functional motor symptoms and there is accumulating evidence for its efficacy and safety from uncontrolled studies and five randomised controlled trials (RCTs).[10–15] These studies used divergent methods and so the optimal protocol is presently unclear, for example, whether to use single pulse (spTMS) or repetitive transcranial magnetic stimulation (rTMS); which brain region to target; how many sessions are needed; and the optimal control intervention. Previous studies have generally found post-intervention functional motor symptom improvements following stimulation of primary motor cortex (M1).[11–15] However, few of these RCTs reported gains in other important outcomes (eg, comorbid psychological/physical symptoms, quality of life/global functioning, health-care resource use). Despite post-treatment improvements in core FND symptoms following rTMS to M1, Taib *et al*,[14] for example, did not observe superior improvements in health-related quality of life (SF-36 vitality/general health) for active rTMS relative to sham-TMS, and no improvements were observed in psychological symptoms. Similarly, McWhirter *et al*[15] reported improvements in subjective symptoms immediately following spTMS of M1 relative to standard care, but no associated improvements in self-reported mental or physical health (SF-12) or clinician-rated disability (Modified Rankin Scale).

Further research is therefore needed to optimise both TMS treatment and RCT protocols to enable more definitive testing of the efficacy of TMS in improving functional motor symptoms, as well as other important outcome domains.[16 17]

## OBJECTIVES

We aimed to explore the feasibility and acceptability of a novel, controlled spTMS protocol for functional limb weakness to inform the design and implementation of a subsequent larger-scale RCT. The protocol consisted of a minimal 'dose', consisting of two brief sessions of spTMS to M1, with the target region tailored to the specific limb weakness reported by each patient. We compared active stimulation delivered above resting motor threshold (RMT) to a control condition consisting of exactly the same procedures delivered below RMT. We hypothesised that this protocol would be feasible in terms of the following key parameters: recruitment rates, acceptance of randomisation, tolerance of the intervention, successful blinding and completion of outcome measures. We also aimed to estimate the variability of outcome measures and treatment effect sizes to inform design of the next RCT.

## METHODS

### Trial design

The study was a double-blind two parallel arm controlled feasibility RCT of tailored spTMS to M1 in patients with functional limb weakness. The primary outcome was patient-rated symptom change. We also measured a range of other relevant secondary outcome domains to assess their feasibility and acceptability in this population (see 'Outcome measures').

### Study setting and participants

Patients with functional limb weakness were recruited from inpatient and outpatient neurology and neuropsychiatry services across the King's Health Partners (National Health Service, UK), including King's College Hospital, Guy's and St Thomas' Hospital, and the South London and Maudsley NHS Foundation Trusts. Recruitment took place between October 2017 and March 2018.

Inclusion criteria were as follows:

► Diagnostic and Statistical Manual of Mental Disorders - Fifth Edition diagnosis of functional neurological disorder confirmed by a consultant neuropsychiatrist or neurologist.
► Motor symptoms defined by functional weakness of at least one limb.
► 18 years old or older.
► Capacity to consent.

Exclusion criteria were as follows:

► Epilepsy (or considered high risk of epilepsy from medical history).
► Other contraindication to TMS (eg, cochlear implants, metallic intracranial clips or intracranial surgery in last 12 months).
► Comorbid neurological condition (eg, multiple sclerosis, stroke).
► Pain as primary symptom.
► Previous treatment with TMS (for any condition).
► Non-fluent English speakers (if unable to accurately complete self-report questionnaires).
► Major mental health disorder: current diagnosis of schizophrenia or bipolar disorder; current drug/alcohol dependence.
► History of factitious disorder.
► Currently involved in another trial.

Preliminary eligibility screening was completed by clinical neurology and neuropsychiatry staff. When patients were considered potentially eligible, Participant Information Sheets were provided (online supplemental file 1), and permission was sought for the research team (TN/SP) to contact the patient. When permission was granted, a member of the research team subsequently contacted the patient to answer any questions and arrange an initial screening assessment visit, if the patient wished to enrol. Written informed consent was obtained at the initial screening visit, after the study had been explained in full and any remaining questions answered. All participants were told that TMS had shown promising results in previous small-scale research studies and that the current

study was aiming to test the treatment more stringently. Hypotheses regarding the possible mechanisms of treatment were not disclosed. Possible side effects of the treatment were outlined (eg, headaches, scalp tingling).

Participants were not reimbursed for involvement in the study, but assistance with travel arrangements and expenses was provided, as necessary.

## Patient and public involvement

A specialist service user advisory group was set up to inform the design and conduct of the study. Key national and international patient groups are involved in the dissemination plans.

## Background/screening measures

At the initial screening visit, demographic details and medical history were obtained and a formal psychiatric screening tool was administered (MINI International Neuropsychiatric Interview).[18] Additional background measures were administered, including a personality disorder screen (Standardised Assessment of Personality–Self-Report),[19] a measure of estimated intellectual functioning (National Adult Reading Test)[20] and a trauma inventory (Childhood Experiences of Care and Abuse Questionnaire).[21]

## Intervention

Participants were randomised to receive active or inactive TMS, as described below. Both groups received two TMS sessions, separated by approximately 4 weeks. A formal script was not used during the sessions, but care was taken to have a consistent and neutral approach in terms of patient interactions regarding potential improvements to minimise and standardise placebo effect.

## Active TMS

The active treatment consisted of spTMS delivered to M1 including stimuli above RMT, thereby causing observable movement of the target limb. The target limb was determined for each participant, defined as the weakest limb (ie, arm or leg on either side) that caused most significant functional impairment in daily life. The target limb remained unchanged throughout both treatment sessions. The treatment was delivered in two phases:

### Phase 1: measuring RMT

Single pulses were delivered with a Magstim 200 (Magstim, Whitland, UK) TMS machine either using a circular coil to the area of M1 corresponding to the hand region of both the symptomatic and the non-symptomatic arms, or using a double cone coil to deliver pulses to the M1 area for the legs (for participants with leg weakness only). As double cone coils cannot target left or right legs separately (M1 for both legs are stimulated), the same procedure was repeated twice as if targeting each side individually so that the procedure was the same for legs as it was for arms.

Pulses started at 20% of machine output and increased at increments of 5% until the evoked response (measured by surface electromyography in the first dorsal interosseous of the hand or extensor digitorum brevis of the foot) exceeded 50 mcV in 50% of trials using a standardised protocol, which allows electromyographic detection of RMT at 5%–10% of TMS output, below that which will produce a movement of the limb detectable by the patient.[22] This value was recorded as the RMT. As a variable number of pulses was needed to establish RMT in each patient, further pulses were then delivered at an interval of 5–10s so that a total of 100 stimuli were delivered (50 stimuli to the same region of M1 bilaterally) to ensure that all participants received an equal number of stimuli during this phase.

### Phase 2: suprathreshold (active) TMS

A further 20 pulses, again at an interval of 5–10s, were delivered at 120% of RMT, applied to the region of M1 corresponding to the participant's weakest limb. No deliberate effort was made by the TMS deliverer to draw attention to the movement of the target limb. A total of 120 pulses were delivered during each of the two treatment sessions. The total number of 120 stimuli was adopted because 100 stimuli is the minimum required to reliably measure RMT and an extra 20 stimuli were needed for clear suprathreshold stimulation for therapeutic effect. This number has been recommended in standardised protocols for RMT measurement.[22]

### Inactive (control) TMS

The inactive control treatment consisted of spTMS delivered to M1 that was always below RMT, thereby not leading to observable movement of the target limb. Phase 1 was identical to the procedures outlined above for measuring RMT.

### Phase 2: Submotor threshold (inactive) TMS

A further 20 pulses at 80% of RMT were applied to the region of M1 corresponding to the patient's weakest limb. While this constituted 'real' TMS, these stimuli did not produce any limb movement. Therefore, the key difference between the treatment conditions was whether stimulation was delivered above or below RMT and the initiation of automatic limb movement or not, respectively. As with the active treatment, a total of 120 stimuli were delivered during each TMS session.

## Changes to protocol during trial

The original protocol specified that the second TMS session would follow the first within a narrowly defined period (30±2 days); however, during the course of the trial, it became clear that this was too restrictive and therefore not practicable, so the time period permitted between treatment sessions was extended (TMS session 2 to occur 28–50 days after TMS session 1).

## Outcome measures

Outcome measures were completed before and/or after the first TMS session (baseline), before and/or after the second TMS session and 3 months after the first TMS

session. The primary outcome measure was patient-rated symptom change assessed with the Clinical Global Impression Improvement (CGI-I) scale,[23] given the emerging consensus that patient-rated, subjective symptom improvements are particularly meaningful outcomes in this disorder.[16 17]

A range of secondary outcome measures was also included to assess the feasibility of measuring other relevant outcome domains in this group:

► Outcome-rater and carer assessed symptom change (CGI-I scale).
► Manual muscle testing (MRC strength scale performed by neurologist).
► Dynamometry (if upper limb weakness present).
► Subjective ratings of strength (0%–100%) and weakness (1-5) in the weakest/target limb.
► Somatic symptoms (Patient Health Questionnaire (PHQ)–15).[24]
► Depression (PHQ-9).[25]
► Overall psychological distress (Core Outcomes in Routine Evaluation–10, CORE-10).[26]
► Quality of life (Short-Form Health Survey–36, SF-36).[27]
► Anxiety (Generalised Anxiety Disorder Questionnaire–7 item, GAD-7).[28]
► Disability/physical functioning (Barthel Index/Functional Independence Measure and Functional Assessment Measure (FIM/FAM)).[29 30]
► Social and occupational functioning (Work and Social Adjustment Scale, WSAS).[31]
► Healthcare utilisation (Client Service Receipt Inventory, CSRI).[32]

## Randomisation and blinding

Randomisation occurred after the initial screening visit, once eligibility and consent had been confirmed. Randomisation was carried out online by the King's Clinical Trials Unit at the Institute of Psychiatry, Psychology and Neuroscience, using block randomisation. Computer-generated randomisation was initiated when the trial outcome-rater (SP) entered the initials and date of birth of the participant onto an online system. Randomisation was then conducted automatically and a confidential email with treatment allocation (active or inactive) sent directly to the TMS deliverer (TRN). The outcome-rater (SP) remained blind to treatment allocation throughout the study, as did participants.

After completion of all study visits for each participant, blinding of the outcome-rater and participant was tested with a forced-choice question about which treatment the patient had received (active or inactive). The patient and outcome-rater answered the question independently. At the end of the study, participants were unblinded individually by the principal investigator (TRN) during debriefing, with the outcome-rater absent from the room. The outcome-rater remained blind to treatment allocation until all outcome data analyses were completed by the trial statistician.

## Safety monitoring

Adverse events (AEs) were monitored and recorded at each study visit and reported to the principal investigator (TRN) or Trial Steering Committee as appropriate. Patients were invited to contact the research team at any time during the trial, in case of an AE occurring between visits.

## Statistical analysis

### Sample size determination

Published data on TMS in FND indicate an improvement rate of approximately 10%, although on the basis of uncontrolled data. As spontaneous recovery rates are very low, a 10% improvement rate in the control arm at 1 month would be a conservative figure. From a previous CBT trial in FND,[7] we would expect 30% of eligible patients to decline participation and then 10% to not complete treatment. Hence, with alpha=0.05% and 90% power, to detect an improvement rate of 80% in the active treatment arm relative to 10% in the control (z test between two independent proportions), nine patients would be needed per arm. For 18 patients to complete the study, given a 10% dropout, we would need to randomise 20 participants (30 consented). This allows 10% dropout rate to be assessed with an expected 95% CI of 0% to 24%.

### Feasibility parameters

Data analysis was carried out in R (V.3.2) by the blinded trial statistician (JH) and adopted the intention-to-treat principle. The aims of the analysis were to examine trial feasibility parameters as follows:

► Recruitment, randomisation and loss to follow-up rates.
► Tolerance of treatment, safety, treatment fidelity, participant/outcome-rater blinding and patient satisfaction.
► Estimate treatment effect sizes as potential outcomes of future trials.

The analysis primarily consisted of descriptive statistics to summarise the rates of consent and randomisation of eligible patients, study retention, data quality (ie, completion of outcome measures, missing data) and the acceptability of TMS to the patient population. Participant demographic and clinical characteristics were also described at baseline.

To assess improvement in symptoms, estimates of treatment effect sizes and 95% CIs on the primary outcome measure (patient-rated CGI-I scale) were obtained using Cliff's delta as this scale is ordinal. Cliff's delta is a functional equivalent to Cohen's d for ordinal data, which does not make assumptions about the shape or spread of the distribution. In this analysis, Cliff's delta represents the mean between-group difference of within-group change. The effect size values can be interpreted as reflecting the number of times a value in one distribution (active group) is higher than the value in the other distribution (inactive group). Criteria for interpreting the effect size were given

by Romano *et al*,[33] with delta <0.147 being negligible, delta <0.33 small, delta <0.474 being medium and otherwise large. For the secondary outcomes, descriptive statistics and effect sizes were calculated as appropriate for the type of data. Effect sizes (and 95% CI) for secondary outcomes were presented as Cohen's d or Cliff's delta as appropriate.

## RESULTS
### Sample characteristics
#### Demographics
The demographic characteristics of participants at enrolment to the study are displayed in table 1. The average age in each group was similar and the majority of participants in both groups was female, right-handed, married/cohabiting and most often of white or black British ethnicity. Participants were most likely to report holding an undergraduate degree or vocational qualification. Participants were most often unemployed, but a proportion of patients reported being retired due to ill-health or employed full time.

#### Background/clinical characteristics
Table 2 shows key background and clinical features of participants at entry to the study. The MINI screen identified one patient with possible current psychosis, who was subsequently excluded and referred to appropriate clinical services. In eligible patients, the most common comorbid mental health diagnoses identified were major depressive disorder (n=8, 38%) and post-traumatic stress disorder (n=6, 29%). A larger proportion of the inactive group reported additional FND symptoms (ie, other than limb weakness), relative to the active group. The duration of time since diagnosis was longer for the inactive group, although the duration since symptom onset was similar across groups. A similar proportion of patients in each group reported concurrent interventions at entry to the study and the average number of medications taken was approximately equal. Full details of concomitant treatments are provided in online supplemental file 2. All participants in both groups were taking medication at every time point, with the most common medications being antidepressant, antiepileptic, anxiolytic and analgesic. The second most common intervention received was physiotherapy (outpatient or during inpatient hospital stays). A small proportion of participants received additional input from occupational therapy, psychology, psychiatry, specialist neurorehabilitation or inpatient hospital (general/neurology) services during the trial.

#### Feasibility parameters
Figure 1 displays rates of recruitment, treatment allocation, completion of the study and participants included in the data analysis (Consolidated Standards of Reporting Trials (CONSORT) flow diagram).

**Table 1** Participant demographic characteristics

| | Active TMS (n=10) | Inactive TMS (n=11) |
|---|---|---|
| Age (median, IQR) | 38 (32.5-46.5) | 41 (33.5–51) |
| **Gender** | | |
| Female | 8 (80) | 10 (90.9) |
| Male | 2 (20) | 1 (9.1) |
| **Marital status** | | |
| Single | 5 (50) | 3 (27.3) |
| Cohabiting/Married | 5 (50) | 7 (63.6) |
| Separated/Divorced | 0 (0) | 1 (9.1) |
| **Qualifications** | | |
| None | 0 (0) | 1 (9.1) |
| GCSE (or equivalent) | 4 (40) | 1 (9.1) |
| A levels | 1 (10) | 0 (0) |
| Graduate | 3 (30) | 3 (27.3) |
| Postgraduate | 0 (0) | 1 (9.1) |
| Vocational | 2 (20) | 5 (45.5) |
| **Employment** | | |
| Full time | 1 (10) | 3 (27.3) |
| Part time | 2 (20) | 0 (0) |
| Unemployed | 7 (70) | 4 (36.4) |
| Retired (ill health) | 0 (0) | 4 (36.4) |
| **Handedness** | | |
| Right | 8 (80) | 8 (72.7) |
| Left | 2 (20) | 2 (18.2) |
| Ambidextrous | 0 (0) | 1 (9.1) |
| **Ethnicity** | | |
| White British | 5 (50) | 7 (63.6) |
| Irish | 1 (10) | 0 (0) |
| White and black Caribbean | 0 (0) | 1 (9.1) |
| Mixed | 1 (10) | 0 (0) |
| Black British | 2 (20) | 2 (18.2) |
| Caribbean | 0 (0) | 1 (9.1) |
| Other | 1 (10) | 0 (0) |

GCSE, General Certificate of Secondary Education; IQR, interquartile range; TMS, transcranial magnetic stimulation.

#### Recruitment, attendance and completion
Of 32 potential candidates referred to the study, 22 consented to participate. Of these, 21 were found to be eligible at baseline screening. All 21 eligible patients were randomised and attended the first TMS treatment session. A total of five patients did not attend the second TMS session (active=4; inactive=1), none gave reasons directly related to the intervention (figure 1). At follow-up, two patients did not attend (active=1; inactive=1). Recruitment of the target number of participants (n=20) was completed within 6 months. The final follow-up session

**Table 2** Background/clinical characteristics by treatment group

| | Active TMS (n=10) | Inactive TMS (n=11) |
|---|---|---|
| SAPAS-SR total scores (median, IQR) | 3 (2–4.8) | 3 (2–4) |
| NART estimated IQ scores (median, IQR) | 107 (105–113) | 108 (108–112) |
| Psychiatric comorbidity present (baseline) (n, %) | 6 (60) | 5 (45.5) |
| Other FND symptoms (baseline) (n, %) | 5 (50) | 9 (81.8) |
| Age at FND onset, years (median, IQR) | 35 (28.25–45) | 31 (23.5–48.5) |
| Duration of FND, months (baseline) (median, IQR) | 41 (14.75–63) | 42 (37–107.5) |
| Duration since FND diagnosis, months (baseline) (median, IQR) | 1 (0–5.25) | 12 (0.5–38.5) |
| Number of current medications (median, IQR) | | |
| Baseline | 3 (2.25–11) | 4 (3.5–6) |
| TMS session 1 | 3 (2–11) | 4 (3.5–6.5) |
| TMS session 2 | 7 (2.25–12.5) | 4.5 (3.25–6.5) |
| Follow-up | 3 (2–12) | 5 (3.5–7) |
| Concurrent treatments (n, %) | | |
| Baseline | 10 (100) | 10 (100) |
| TMS session 1 | 10 (100) | 9 (100) |
| TMS session 2 | 6 (100) | 8 (100) |
| Follow-up | 9 (100) | 9 (100) |

FND, functional neurological disorder; IQ, intelligence quotient; IQR, interquartile range; NART, National Adult Reading Test; SAPAS-SR, Standardised Assessment of Personality Abbreviated Scale–Self-Report; TMS, transcranial magnetic stimulation.

took place approximately 9 months after commencement of the study.

## Data quality

For each visit, the percentage return for each outcome measure was calculated in relation to the number of patients who attended that session (online supplemental file 3). Completion rates for the primary outcome measure (patient-rated CGI-I scale) was 100% at all timepoints. For most other measures, return rates were between 90% and 100% (ie, outcome-rater CGI-I scale, Barthel Index, GAD-7, PHQ-9, PHQ-15, WSAS, CORE-10, most SF-36 subscales). A small number of scales were completed less consistently, although rates were still above 80% (eg, SF-36 Role Emotional at TMS session 1, patient strength ratings/dynamometry at follow-up). Two measures were completed infrequently (carer-rated CGI-I scale/FIM-FAM) in 25% or fewer of the attendees at each timepoint.

## Blinding

There were no unexpected compromises to blinding during the study procedures. When asked with a forced-choice question at the end of the study, the active treatment was more likely to be correctly guessed as active by both patients (40%) and the outcome-rater (50%), compared with the inactive treatment (patients=36%; outcome-rater=27%). The percentage of correct responses by either informant was not above chance.

## Patient satisfaction

Patients' ratings of their overall experiences of the trial were good. The majority of patients (76%) stated that they were either 'somewhat' or 'very' satisfied with the trial, although ratings were higher in the inactive group (active=60%, inactive=92%). None of the patients in either group reported being 'unsatisfied' (neither 'somewhat' nor 'very'). Qualitatively, patients reported feeling pleased with the level of support and information provided by the research team, felt valued, found assistance with travel arrangements beneficial and were pleased to be part of a study that could help people with FND more broadly. For some patients, lack of improvement and/or unwanted side effects were noted in the feedback to explain less positive satisfaction ratings (ie, 'neither satisfied nor unsatisfied').

## Adverse events

There were four serious adverse events (SAEs) reported during the study (active=3; inactive=1). One SAE occurred between TMS sessions 1 and 2, and the other three occurred between TMS session 2 and follow-up. There were no SAEs immediately following a TMS session and none of the SAEs were considered related to the treatment by the Trial Steering Committee. In total, there were 78 (non-serious) AEs, with 15 of these occurring before the first treatment session. Following the start of treatment, there were 26 AEs in the active group and 37 in the inactive group. A proportion of patients in each group reported headaches at some

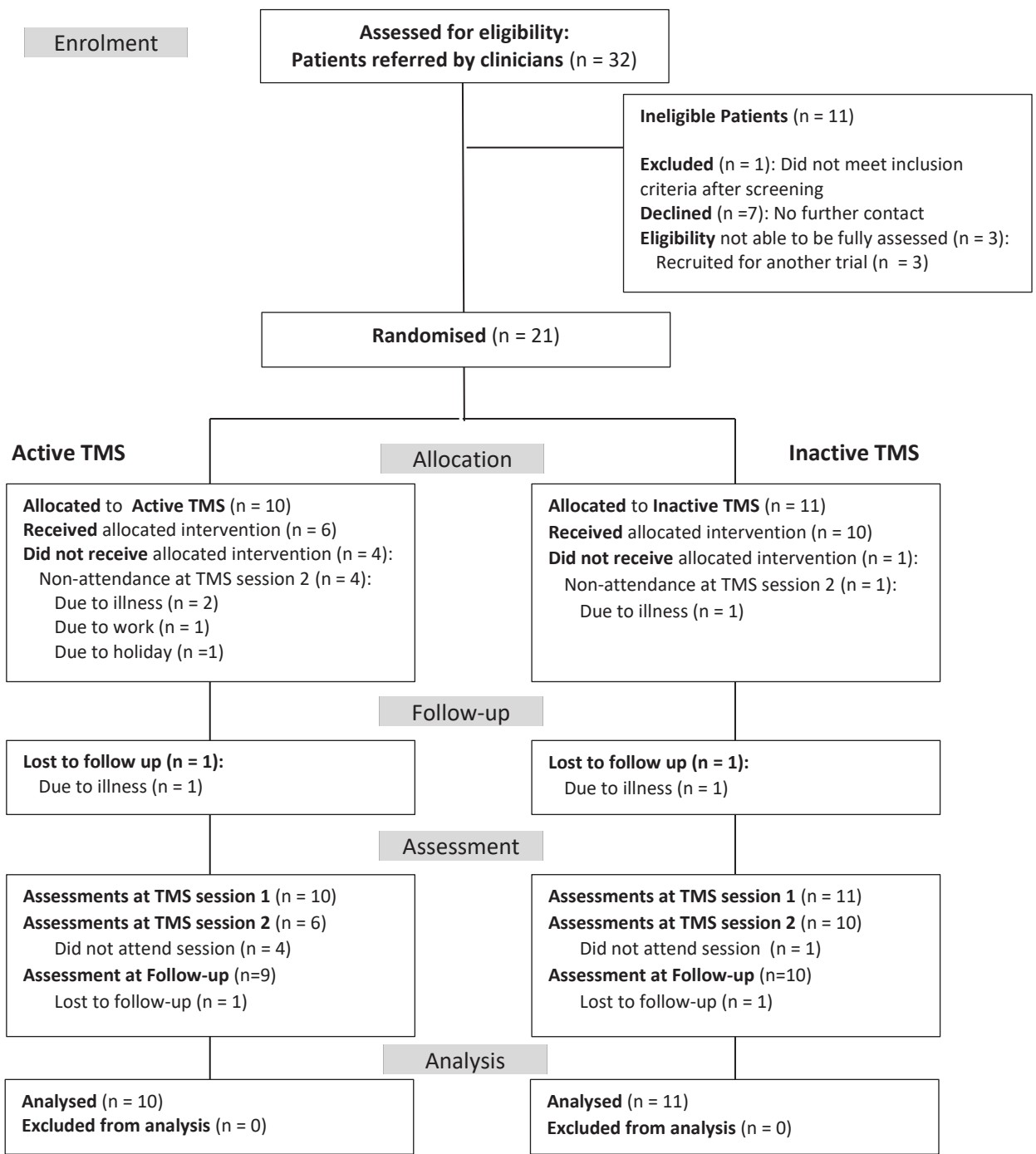

**Figure 1** Consolidated Standards of Reporting Trials diagram. TMS, transcranial magnetic stimulation.

time during the trial (inactive n=5; active n=3). Worsening of FND symptoms was reported by some patients in each group at one or more time point (inactive n=15; active n=12).

### Primary outcome: patient-rated CGI-I scores

Figure 2 displays the patient-rated CGI-I scores by group. Immediately prior to TMS session 1, one participant (9%) in the inactive group and 0% of the active group rated their symptoms as 'much improved' relative to

their condition at entry to the study. Immediately after TMS session 1, these ratings remained the same. Immediately prior to TMS session 2, 67% of patients in the active group and 20% in the inactive group reported that their symptoms were 'much improved'. The relative percentage of 'much improved' again remained the same immediately following TMS session 2. Finally, at 3-month follow-up, the number 'much improved' was 44% in the active group and 20% in the inactive group.

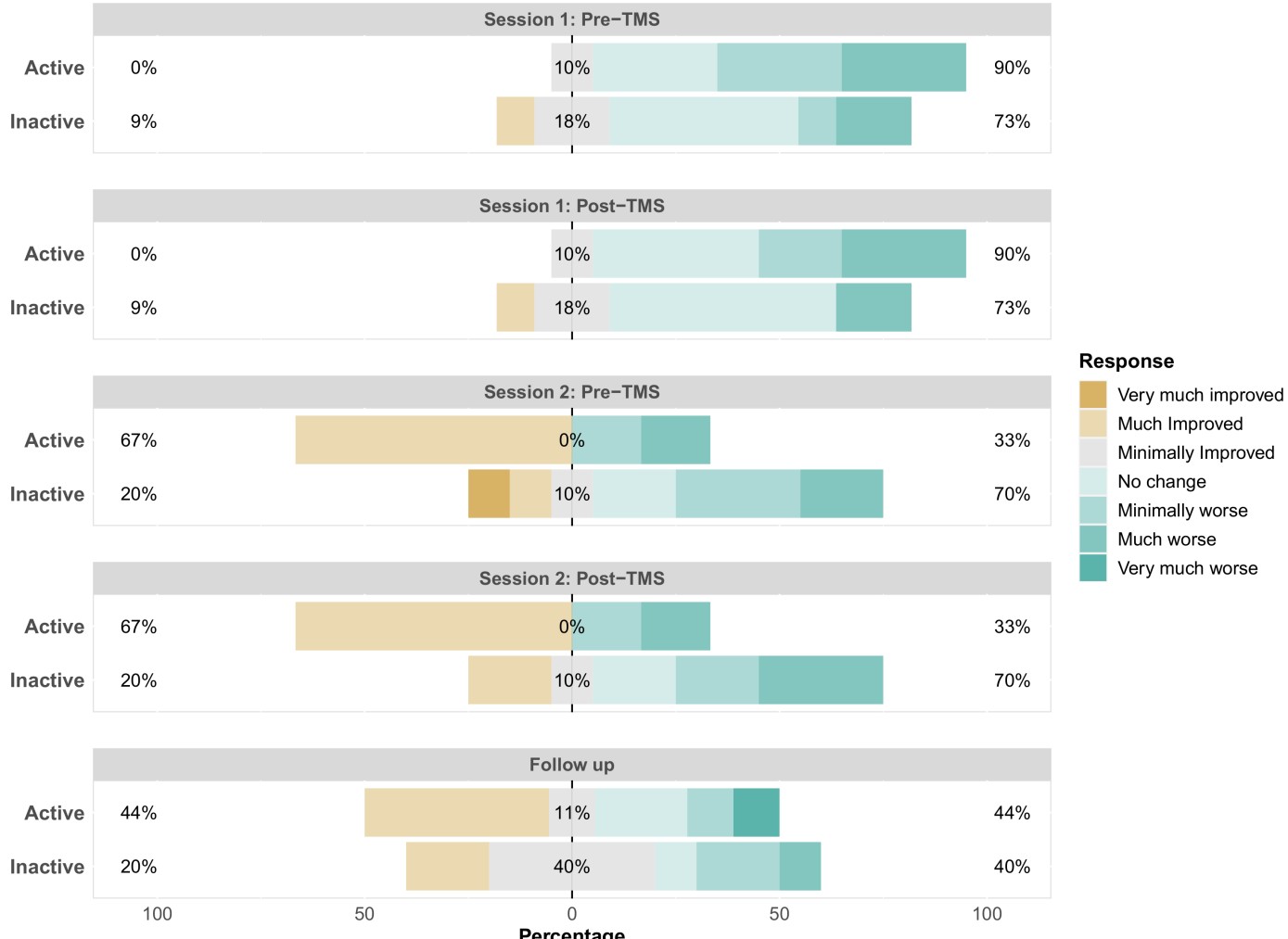

**Figure 2** Patient-rated Clinical Global Impression–Improvement categories by treatment group and timepoint. TMS, transcranial magnetic stimulation.

Effect sizes and 95% CIs (Cliff's delta) for patient-rated CGI-I scores were calculated. The effect size was positive prior to TMS session 1 reflecting coincidentally worse ratings in the active group (Cliff's delta=0.35 (−0.17 to 0.71)). This difference remained the same immediately following TMS session 1 (Cliff's delta=0.35 (−0.15 to 0.7)). However, this pattern was reversed by TMS session 2, indicating a benefit for the active treatment with moderate effect sizes pre-treatment (Cliff's delta=−0.35 (−0.73 to 0.19)) and post-treatment (Cliff's delta=−0.44 (−0.79 to 0.13)). At 3-month follow-up, there was still an advantage for the active treatment; however, the difference was smaller (Cliff's delta=−0.2 (−0.6 to 0.28)), potentially due to a relative improvement in the inactive group.

### Secondary outcomes
Descriptive statistics, effect sizes and CIs for the secondary outcomes can be found in online supplemental file 4. There was considerable variability in the effect sizes and 95% CIs for these outcomes and so the findings cannot be interpreted conclusively. However, the pattern of findings for the following outcomes suggested a benefit of active TMS: outcome-rater CGI-I scores, psychological distress

(CORE-10), aspects of quality of life (SF-36 physical functioning, vitality/energy, role limitations due physical and emotional factors), activities of daily living (Barthel), primary care service use. The following outcomes did not suggest a benefit of active TMS: grip strength (dynamometry), subjective (patient-rated) limb strength, additional physical symptoms (PHQ-15), anxiety (GAD-7), depression (PHQ-9), some aspects of quality of life (SF-36 bodily pain, social functioning, mental health), social/occupational functioning (WSAS), inpatient hospital admissions and total outpatient healthcare contacts.

### DISCUSSION
This novel double-blind RCT of spTMS to M1 for the treatment of functional limb weakness was found to be feasible in terms of key parameters, allowing estimation of the effect sizes for key outcome variables, and to inform the planning and implementation of a larger RCT.

### Feasibility
Rates of recruitment and retention were acceptable, with only two patients (10%) failing to complete the follow-up

visit. While five patients did not attend TMS session 2, none of these instances was directly related to the nature of the intervention. Nevertheless, consideration should be given to ways of improving attendance rates at the second TMS session, such as offering the session earlier (eg, after 1 or 2 weeks) and ensuring that any barriers to attendance are identified and managed in advance.

Completion of outcome measures was generally good, with rates of 90%–100% for most scales. However, the carer-rated CGI-I scale and the FIM-FAM were not completed frequently. Reasons for the lack of completion of the carer-rated CGI-I related to carers not being present or different carers attending each appointment. In the future, a specific carer could be identified at the start of the study (in consultation with the patient) and ratings could be obtained by telephone, should that carer be absent at specific visits. It became clear that the FIM-FAM was not a suitable measure for this study because it requires completion on an inpatient basis, usually by one or more members of a multidisciplinary clinical team. In this study, patients were recruited from a range of outpatient and inpatient settings, and ratings from inpatient clinical teams were at times difficult to obtain. Furthermore, several items on the measure replicated similar constructs assessed within other measures used in the trial (ie, Barthel, SF-36).

Blinding appeared to be successful, with correct identification of active treatment below chance for both the patients and the outcome-rater at the end of the study. Patient satisfaction ratings were also encouraging, suggesting that the trial procedures and the intervention were acceptable in this population. There were no SAEs directly related to the intervention and rates of potentially related AEs (ie, headaches, FND symptom worsening) were not reported at higher rates in the active group. AEs should be closely monitored in future studies.

## Outcomes
### Primary outcome: patient-rated symptom improvement
Point estimates for the patient-reported symptom improvement showed superiority for the active spTMS intervention relative to the inactive intervention, with small to moderate effect sizes. Improvements were most apparent at TMS session 2 but were still evident at follow-up. It is notable that the pattern of scores on the outcome-rater CGI-I scale concurred with the patient-rated CGI-I scores. These findings suggest that tailored spTMS, delivered above RMT to the area of M1 corresponding to a target limb (ie, that limb which is functionally weakest) and thus causing movement of that limb, potentially could lead to greater improvements than the same intervention delivered below RMT (ie, not inducing observable movement). These results concur with those of other studies[11–15] which have previously shown improvements in subjective or objective measures of functional motor symptom severity following spTMS or rTMS to M1.

The mechanism by which TMS to M1 yields improvements in functional motor symptoms is unclear. It is possible that a neuromodulatory mechanism may operate in protocols using rTMS and/or that a general placebo effect could be responsible for improvements in cases where patients/outcome assessors are not blind to treatment allocation. However, similar to Garcin et al,[12] our study suggests that elicitation of normal function of the weak limb with minimal doses of spTMS is sufficient to induce improvements, at least in the short term. Induction of observable normal function in the limb might result in modification of patients' beliefs and expectations regarding limb functioning and the possibility of recovery, and/or may represent a form of motor retraining effect. It is notable that the improvements did not occur immediately after the first treatment but were instead evident by the second treatment session (pre-TMS), suggesting that while one TMS session was sufficient to induce change, the mechanism by which change occurred required time to manifest as symptom reduction.

The findings in this study suggest that the patient-rated CGI-I scale is acceptable and sensitive to change as a measure of symptom improvement in FND intervention studies, in accordance with previous findings across treatment modalities and FND symptom types. This measure has recently been recommended as a primary outcome measure in FND treatment studies.[17]

### Secondary outcomes
High rates of completion of most of the secondary outcome measures indicated that they are appropriate tools for use in future, similar studies. Of the range of outcome domains included, the clearest trends for intervention-related improvements were in activities of daily living/disability (Barthel), overall psychological distress (CORE-10), aspects of health-related quality of life (ie, physical functioning, physical role, vitality, emotional role) and primary care service use. While extreme caution should be exercised in interpreting these findings due to the small sample size, small-moderate effect sizes and variable confidence intervals, these initial findings suggest that active spTMS might be associated with improvements in aspects of mental health, daily functioning (ie, roles, daily activities, physical) and treatment seeking, in addition to core FND symptom improvements. This extends the findings of previous studies, which have generally demonstrated improvements in functional motor symptoms only. However, it is not possible to say whether improvements in these additional outcome domains followed or preceded motor symptoms.

### Strengths and limitations
A key strength of this study included the use of a minimal TMS protocol (two brief sessions of spTMS only), which was acceptable to patients and therefore resulted in good treatment adherence rates. This minimal TMS protocol also has potential to be used as a widely accessible treatment that could be adjunctive to other therapies in a range of settings.

Another strength was that our inactive intervention was similar enough to the active treatment (ie, 'real' TMS) to reduce the risk of patients inadvertently becoming unblinded to treatment allocation. Furthermore, blinding of both patients and outcome assessors ensured that post-treatment gains were not due entirely to general placebo effects. The inclusion of patients with additional functional neurological symptoms, non-major psychiatric comorbidities and those undergoing concomitant treatments yielded a sample that was representative of the broader FND patient population, improving the generalisability of the findings.

However, it is possible that the additional interventions that some patients were undergoing (eg, physiotherapy, specialist neurorehabilitation) may have facilitated some of the improvements reported following treatment. Future RCTs with larger samples should balance the influence of concomitant treatments and/or any incidental baseline between-group differences in symptoms, background features or other relevant variables.

Another limitation to note is that some degree of improvement in FND symptoms was observed in both groups prior to commencing the first TMS session, relative to enrolment to the study. It is therefore unclear whether the improvements observed following TMS reflected the effect of the intervention (including its anticipation) or the natural course of the disorder. The lack of a formal script during treatment sessions might have led to inconsistencies in placebo effect. Future studies might valuably include an additional standard care or waiting-list control group to examine these factors.

## CONCLUSION

The findings suggest that active (supra-motor threshold) spTMS to M1 is a safe, efficient, acceptable and potentially effective treatment for functional limb weakness, leading to improvements in core symptoms and potentially other important outcome domains. A larger, pilot RCT is now warranted to obtain a more robust estimate of effect sizes and variability in outcomes for this promising intervention.

**Author affiliations**
[1]Institute of Psychiatry Psychology and Neuroscience, King's College London, London, UK
[2]Department of Neurology, King's College Hospital NHS Foundation Trust, London, UK
[3]Department of Clinical Neurophysiology, King's College Hospital NHS Foundation Trust, London, UK
[4]Institute of Mental Health, University College London, London, UK

**Contributors** TRN, AP and ASD developed the study design. TRN wrote the ethics proposal/study protocol, recruited some participants and conducted the TMS sessions. SP recruited and screened participants, conducted baseline and all subsequent outcome assessments, cleaned/entered data, and wrote the first/subsequent drafts of the manuscript. JH conducted the statistical analyses, prepared the CONSORT flow diagram and some sections of the results. BS, KS, JB, HA, IS, and AE conducted clinical strength tests during outcome assessments. All authors contributed to editing of the manuscript for important intellectual content and approved the final version prior to submission.

**Funding** TRN, SP and the study were funded by a National Institute of Health Research (NIHR) Clinician Scientist Award to TRN. This article represents independent research funded by the NIHR. The views expressed are those of the authors and not necessarily those of the NHS, NIHR or the Department of Health and Social Care.

**Competing interests** None declared.

**Patient consent for publication** Not required.

**Ethics approval** The study was reviewed and approved by the London-Stanmore NHS Research Ethics Committee, study reference number 17/LO/0410). All participants provided informed, written consent prior to involvement in the study.

**Provenance and peer review** Not commissioned; externally peer reviewed.

**Data availability statement** All data relevant to the study are included in the article or uploaded as supplemental information.

**ORCID iDs**
Susannah Pick http://orcid.org/0000-0003-2001-6723
John Hodsoll http://orcid.org/0000-0001-7546-9901
Biba Stanton http://orcid.org/0000-0003-2275-9109
Amy Eskander http://orcid.org/0000-0001-5527-9801
Ioannis Stavropoulos http://orcid.org/0000-0002-6920-1826
Kiran Samra http://orcid.org/0000-0002-3105-7099
Anthony S David http://orcid.org/0000-0003-0967-774X
Alistair Purves http://orcid.org/0000-0002-4805-4417
Timothy R Nicholson http://orcid.org/0000-0002-4805-4417

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
