## [Reviewer comments · BMJ Open]

ARTICLE DETAILS

TITLE (PROVISIONAL)	Trial Of Neurostimulation In Conversion Symptoms ('TONICS'): a feasibility randomised controlled trial of transcranial magnetic stimulation for functional limb weakness
AUTHORS	Pick, Susannah; Hodsoll, John; Stanton, Biba; Eskander, Amy; Stavropoulos, Ioannis; Samra, Kiran; Bottini, Julia; Ahmad, Hena; David, Anthony; Purves, Alistair; Nicholson, Timothy

VERSION 1 – REVIEW

REVIEWER	Laura McWhirter University of Edinburgh, UK
REVIEW RETURNED	06-Mar-2020

GENERAL COMMENTS	This pilot study of TMS as a treatment for functional limb weakness was interesting to read and the participants and technical TMS protocol were clearly described. I particularly appreciated the thoughtful discussion about selection of appropriate outcome measures: a good demonstration of how helpful the authors' other work on outcome measures will be for FND treatment trials in the future. There are two areas where I think more information would be helpful. Firstly, the authors recognise the potential confounding effect of other treatments between treatment 1 and 2, and I see that almost all participants were having concurrent treatments. It would be helpful to know about the nature and intensity of these other treatments. Secondly, I think it is very important here to explain a) what the participants were told about the treatment and how it might work before the treatment started, and b) what happened to participants during treatment in terms of any verbal suggestion. The role of suggestion both verbal and related to the manner of delivery might be an important part of the effect of TMS. In my view this is as important a part of the protocol as, for example, the method of measuring motor threshold (which is well described). If no script or attempt to standardise suggestion was used this might be mentioned as a potential limitation. I enjoyed reading the paper and look forward to the results of the proposed RCT.
--

REVIEWER	Matthew J. Burke
-----------------	------------------

	University of Toronto, Canada
REVIEW RETURNED	07-Mar-2020

GENERAL COMMENTS	Pick et al. conduct an important randomized pilot trial of TMS for the treatment of functional weakness. Unfortunately, their study has major methodological concerns. The authors make many references to this being a placebo-controlled study. The comparator group in this study is NOT placebo or sham. Applying TMS at 80% of RMT (their “placebo” group) has been shown to modulate cortical excitability and is the intensity used in most TMS neuromodulation treatment studies (eg for FDA-approved depression treatment protocols). What the authors are essentially comparing is demonstration of limb movement using active TMS (above RMT) versus no demonstration of limb movement using active TMS (below RMT). The wording to reflect this needs to be changed throughout the entire manuscript. More emphasis, rationale and discussion need to be placed on the potential importance of this contrast (as demonstration of limb movement could indeed be a critical factor for why FND patients improve after TMS). A further complicating issue is that the process of determining RMT leads to demonstration of limb movement (hand jerks accompanying the MEPs). Unless, I was mistaken reading the methods, it seems that both groups received 100 pulses for identifying the RMT. Thus, the comparator group also had some demonstration of jerks...with the only major difference between the groups being the additional 20 pulses being below or above RMT. Further discussion of appropriate sham controls in TMS studies has been described elsewhere: Davis NJ, Gold E, Pascual-Leone A, Bracewell RM. Challenges of proper placebo control for non-invasive brain stimulation in clinical and experimental applications. European Journal of Neuroscience. 2013 Oct;38(7):2973-7. As the authors allude to and as discussed previously (Pollak et al 2014, JNNP; Burke et al 2018, Mov Disord Clin Pract), there are three main mechanisms in which TMS for FND is posited to potentially work. 1) TMS-induced neuromodulation of the relevant brain network (in this case motor), 2) a cognitive/behavioral effect (showing the patient that their limb can move) or 3) placebo effects. These mechanisms are not necessarily mutually exclusive. To isolate and test the importance of that second potential factor in a randomized trial would be very valuable for the field. However, the manuscript would have to be revised considerably to reflect this and the concern still exists that both groups may have received demonstration of limb movement (during the RMT identification phase). Finally, the findings of safety/tolerability of TMS in this patient population are important and are not impacted by the above concerns.
--

REVIEWER	Deepak Nag Ayyala Medical College of Georgia, Augusta University, USA
REVIEW RETURNED	21-Apr-2020

GENERAL COMMENTS

The study is aimed to explore the feasibility and acceptability of a novel, placebo-controlled spTMS protocol for functional limb weakness. The primary outcome of the RCT was patient-rated symptom change and the secondary outcomes included clinician-rated symptom change, psychosocial functioning and disability. The outcome measures are clearly defined in the abstract and the objectives. However, there are a few concerns that need to be addressed with regards to the study design and the statistical analysis of the study data.

Major concerns

1. The authors define the primary outcome as the patient-rated symptom change, which is measured as an improvement rate in the sample size determination. However, assessment of improvement in symptoms is done using the Cliff's Delta as the measure. These are different measures are the former measures the within-treatment improvement in symptoms and the latter measures the between-treatments improvement in symptoms. The authors need to provide how the improvement rate has changed within each treatment and then compare them to justify their usage for the sample size analysis for the trial.
2. The power analysis seems inaccurate. The goal of the study is to see if the active arms has enhanced improvement rate compared to the inactive TMS arm. As such a one-tailed power analysis gives a sample size of ($n = 7$) as opposed to a two-tailed power analysis (with $n = 9$) as reported in the manuscript. While the study would still be powered enough since a larger sample size is used, the sample size mentioned needs to be changed.
3. In the analysis of adverse events, the authors make two claims: (a) rates of reported headaches were higher in the inactive group (5 out of 10) relative to the active group (3 out of 11) (b) worsening of FND symptoms was higher in the inactive group (15) versus the active group (12). These claims are misleading and should be substantiated by a comparative analysis. For (a), the difference between the groups is not significant (p-value of 0.2672 with a CI of (-0.21,1) for the difference of rates of reported headaches). The lack of difference between the inactive and active groups should be clearly presented.
4. Referring to my first comment, I am concerned about the lack of comparison between the two groups at any time point in the study. For example, consider the patient ratings. Within the active arm, the Cliff's delta for visit 1(pre-TMS) vs. visit 2(pre-TMS) is 0.53 (large). However, the Cliff's delta for visit 2(pre-TMS) vs. follow-up(pre-TMS) is -0.12 (negligible). A similar trend can be observed for the inactive arm, with the second visit having a "small" Cliff's delta. How does this justify the need for a second visit when the ratings have not significantly changed? Maybe a RCT with a single visit is sufficient to observe a

	significant improvement in the patient ratings, with the second visit not having a significant improvement. Minor concerns  1. Can the inactive arm be called a true “placebo”? Or is it standard of practice? 2. In the sample size analysis, the authors mention that in “a previous CBT trial in FND”, they expect 30% of consented patients to decline randomization. If this observation is from a published work, a reference needs to be provided. Otherwise, it must be clearly stated as an unpublished result. 3. Which language/software was used to perform the statistical analysis? R? SAS? This information should be provided along with the version used.
--	--

VERSION 1 – AUTHOR RESPONSE

Reviewer: 1. Laura McWhirter

There are two areas where I think more information would be helpful.

Firstly, the authors recognise the potential confounding effect of other treatments between treatment 1 and 2, and I see that almost all participants were having concurrent treatments. It would be helpful to know about the nature and intensity of these other treatments.

Response: We agree with this point and have added more information on concomitant treatments on p.16 and in Supplementary File 2.

Secondly, I think it is very important here to explain a) what the participants were told about the treatment and how it might work before the treatment started, and b) what happened to participants during treatment in terms of any verbal suggestion. The role of suggestion both verbal and related to the manner of delivery might be an important part of the effect of TMS. In my view this is as important a part of the protocol as, for example, the method of measuring motor threshold (which is well described). If no script or attempt to standardise suggestion was used this might be mentioned as a potential limitation.

Response: We also agree with this set of inter-related points and have added more information on what was explained about the treatments in advance in terms of the Patient Information Sheets (pp.7-8) and explanations given by the research team during sessions (pp.8-9). A formal/strict script was not used for the treatment sessions but the same member of the research team (TRN) was present for all treatment sessions to ensure neutral language (and tone) was used similarly for all patients at both treatment sessions for the reasons the reviewer suggested. The lack of script has been added to the methods section and mentioned as a potential limitation (p.27).

Reviewer: 2 Matthew J. Burke

Unfortunately, their study has major methodological concerns. The authors make many references to this being a placebo-controlled study. The comparator group in this study is NOT placebo or sham. Applying TMS at 80% of RMT (their “placebo” group) has been shown to modulate cortical excitability and is the intensity used in most TMS neuromodulation treatment studies (eg for FDA-approved depression treatment protocols). What the authors are essentially comparing is demonstration of limb movement using active TMS (above RMT) versus no demonstration of limb movement using active TMS (below RMT). The wording to reflect this needs to be changed throughout the entire manuscript. More emphasis, rationale and discussion need to be placed on the potential importance of this contrast (as demonstration of limb movement could indeed be a critical factor for why FND patients improve after TMS).

Response: We partially agree with this statement. We agree that in our study design the demonstration of limb movement is a key aspect of what we think are the potential ‘active ingredients’ of TMS in this context, although it could equally be that moving ‘in and of itself’ is critical in terms re-experiencing and relearning movement and it is possible there are several other factors such that it might not just be the ‘demonstration’ of movement per se. This is why we designed this trial in the way we did – so as to make the inactive treatment the same in as many aspects as we could in order to be best placed to infer the reason for any differences between the groups are due to the proposed ‘active ingredient(s)’.

Base on personal communications with international neurophysiology experts Prof Kerry Mills and Prof Mark Hallett the single pulses of TMS delivered every 5-10 seconds in this design for inactive/control arm are to not likely to cause significant neuromodulation of the motor cortex/system (although we concede this has not to their knowledge been formally studied) compared to the long trains of high frequency ‘repetitive’ TMS referred to in depression treatment protocols. Therefore, we felt it was justifiable to call our control intervention ‘placebo’. Nevertheless, in light of these comments we have removed references to placebo throughout and instead refer only to a ‘control’ intervention.

We decided to not make explicit reference to the mechanism of demonstrating movement (or experiencing/other aspects of movement) in the paper (or the publicly visible protocol on ISRCTN) as this would potentially unblind participants in the next larger (pilot) phase study if they read this.

A further complicating issue is that the process of determining RMT leads to demonstration of limb movement (hand jerks accompanying the MEPs). Unless, I was mistaken reading the methods, it seems that both groups received 100 pulses for identifying the RMT. Thus, the comparator group also had some demonstration of jerks...with the only major difference between the groups being the additional 20 pulses being below or above RMT.

Response: After consultation with neurophysiology/TMS experts as part of the planning of this study (initially Professor Kerry Mills, whose work pioneered RMT definitions and detection methods, and then his successor as head of the department, and co-author on this paper, Dr Alistair Purves) we designed this study using the (See Mills et al, ref 22 in the manuscript) protocol for RMT determination which usually allows EMG detection of RMT 5-10% of TMS output below that which will produce a movement of the limb detectable by the patient. We agree that such movement felt by the patient in the control arm of the study would mean they could no longer be in the control arm and would need to exit the study. We have added this important detail to the methods section of the paper (pp.9-10).

Further discussion of appropriate sham controls in TMS studies has been described elsewhere: Davis NJ, Gold E, Pascual-Leone A, Bracewell RM. Challenges of proper placebo control for non-invasive brain stimulation in clinical and experimental applications. European Journal of Neuroscience. 2013 Oct;38(7):2973-7.

Response: We agree that this is a valuable paper on the challenges of placebo/control group in TMS trials, as are more recent papers covering the latest sham devices, but feel on balance that a detailed discussion of this complex issue is beyond the scope and word limit of this paper.

As the authors allude to and as discussed previously (Pollak et al 2014, JNNP; Burke et al 2018, Mov Disord Clin Pract), there are three main mechanisms in which TMS for FND is posited to potentially work. 1) TMS-induced neuromodulation of the relevant brain network (in this case motor), 2) a cognitive/behavioural effect (showing the patient that their limb can move) or 3) placebo effects. These mechanisms are not necessarily mutually exclusive.

To isolate and test the importance of that second potential factor in a randomized trial would be very valuable for the field. However, the manuscript would have to be revised considerably to reflect this and the concern still exists that both groups may have received demonstration of limb movement (during the RMT identification phase).

Response: These concerns are hopefully adequately addressed above.

Reviewer: 3 Deepak Nag Ayyala

The study is aimed to explore the feasibility and acceptability of a novel, placebo-controlled spTMS protocol for functional limb weakness. The primary outcome of the RCT was patient-rated symptom change and the secondary outcomes included clinician-rated symptom change, psychosocial functioning and disability. The outcome measures are clearly defined in the abstract and the objectives. However, there are a few concerns that need to be addressed with regards to the study design and the statistical analysis of the study data.

Major concerns

1. The authors define the primary outcome as the patient-rated symptom change, which is measured as an improvement rate in the sample size determination. However, assessment of improvement in symptoms is done using the Cliff's Delta as the measure. These are different measures are the former measures the within-treatment improvement in symptoms and the latter measures the between-treatments improvement in symptoms. The authors need to provide how the improvement rate has changed within each treatment and then compare them to justify their usage for the sample size analysis for the trial.

Response: The trial is a parallel group design with the key comparison between groups at designated time-points. Cliff's delta represents the mean between group difference of within group change (as measured by the CGI-I). We have this detail on p.15.

2. The power analysis seems inaccurate. The goal of the study is to see if the active arms has enhanced improvement rate compared to the inactive TMS arm. As such a one-tailed power analysis gives a sample size of (n = 7) as opposed to a two-tailed power analysis (with n = 9) as reported in the manuscript. While the study would still be powered enough since a larger sample size is used, the sample size mentioned needs to be changed.

Response: As is standard for an RCT, we entered into the trial in a state of clinical equipoise, meaning we didn't know whether the treatment would be beneficial or not. As there might be a positive or negative impact of treatment a two-tailed test is appropriate. We, therefore, have not altered our power calculation.

In the analysis of adverse events, the authors make two claims: (a) rates of reported headaches were higher in the inactive group (5 out of 10) relative to the active group (3 out of 11) (b) worsening of FND symptoms was higher in the inactive group (15) versus the active group (12). These claims are misleading and should be substantiated by a comparative analysis. For (a), the difference between the groups is not significant (p-value of 0.2672 with a CI of (-0.21,1) for the difference of rates of reported headaches). The lack of difference between the inactive and active groups should be clearly presented.

Response: Our description of adverse events is descriptive rather than based on inference from formal hypothesis tests. These are not appropriate in a feasibility trial and particularly with a sample size of 10 per group. As the trial was not powered to target these differences, it is not possible to draw any conclusions from these p-values.

3. Referring to my first comment, I am concerned about the lack of comparison between the two groups at any time point in the study. For example, consider the patient ratings. Within the active arm, the Cliff's delta for visit 1(pre-TMS) vs. visit 2(pre-TMS) is 0.53 (large). However, the Cliff's delta for visit 2(pre-TMS) vs. follow-up(pre-TMS) is -0.12 (negligible). A similar trend can be observed for the inactive arm, with the second visit having a "small" Cliff's delta. How does this justify the need for a second visit when the ratings have not significantly changed? Maybe a RCT with a single visit is sufficient to observe a significant improvement in the patient ratings, with the second visit not having a significant improvement.

Response: There are currently no data on the optimal number of TMS sessions required, but based on our review of the existing literature and clinical experience (that more than one session, ideally quite soon after the first, can often be needed before clinical change occurs) we predicted that two sessions would be optimal and practicable for this study.

As the focus of our study is trial feasibility (i.e. is it possible to do the trial, recruitment, retention and patient experience), we reported ES (and 95% CI) as is standard but given the small sample size, large sampling (and ES) variability would be expected. On the decision to include a second visit, we would base this on feasibility considerations rather than inferential analysis of data at hand, which is not possible given the small sample size.

Minor concerns

1. Can the inactive arm be called a true “placebo”? Or is it standard of practice?

Response: We hope this is adequately addressed in the response to reviewer 2

2. In the sample size analysis, the authors mention that in “a previous CBT trial in FND”, they expect 30% of consented patients to decline randomization. If this observation is from a published work, a reference needs to be provided. Otherwise, it must be clearly stated as an unpublished result.

Response: We have added this reference

3. Which language/software was used to perform the statistical analysis? R? SAS? This information should be provided along with the version used.

Response: Statistical analysis was carried out in R 3.2. We have added this to the methods section (p.14).

VERSION 2 – REVIEW

REVIEWER	Laura McWhirter University of Edinburgh
REVIEW RETURNED	03-Jul-2020

GENERAL COMMENTS	This paper describes a feasibility trial of 2 sessions of single pulse TMS to the primary motor cortex at above vs below motor threshold as a treatment for functional motor disorder (here, limb weakness). The choice of outcome measures is clearly justified and the TMS protocol is clearly described. Although described as a feasibility trial, numbers in previous trials of TMS for functional disorder are small, and this is a valuable addition to the evidence in this area. As placebo factors and suggestion may be important in the efficacy of TMS, ideally a script would be used to attempt to standardise these elements and ensure reproducibility, especially as the person delivering treatment is necessarily not blinded to treatment arm. Regarding the trial itself, one possible confounder is that many participants were simultaneously receiving other psychotherapy or neurorehabilitation treatments. Again, a future trial might standardise these methods. These limitations are adequately acknowledged by the authors. My only suggestion for revision regards figure 2 - I think the labelling of this figure could be improved as it takes some scrutiny to learn what each of the percentages inside the table indicate. I also wonder if treatment arm could be labelled directly rather than using a key.
--

REVIEWER	Matthew Burke, MD FRCP University of Toronto, Canada
REVIEW RETURNED	University of Toronto, Canada 23-Jun-2020

GENERAL COMMENTS	I appreciate the authors' responses to my review. Thank you for the clarification of the method used to determine RMT - including that muscle twitches/jerks were not being elicited in this process (re: concern with control group). As mentioned by the authors in their response, it is not possible to definitively know (has not been previously studied) whether low frequency single TMS pulses at 80% RMT (as administered to the control group) may be able to modulate cortical excitability relevant to dysfunctional motor networks in FND. These types of pulse parameters have been shown to transiently modulate the motor cortex (eg cortical inhibition and facilitation protocols) but agree this is different from the high frequency rTMS protocols used in depression treatment. I had sent an addendum to the editors that the latter was mistakenly referenced in the initial review and should have read "TMS neurophysiological studies (eg cortical inhibition and facilitation protocols)". I am happy to see that the comparison group has no longer been described as a placebo group given the above discussion. The authors have adequately addressed my concerns. This study offers a meaningful contribution to the limited literature on this topic.
---

REVIEWER	Deepak Nag Ayyala Medical College of Georgia, Augusta, Georgia, USA
REVIEW RETURNED	06-Jul-2020

GENERAL COMMENTS	The authors have clearly answered most of the concerns in the previous review. I'd still want the authors to clarify a few points with regards to their responses. 1. For the analysis of the adverse events, the authors responded saying the study is not powered to target the differences between active and inactive groups. In that case, their presentation of the result as "Worsening of FND symptoms was reported by some patients in each group at one or more time point, but the frequency of such reports was higher in the inactive group (15) compared to the active group (12)." sounds to be strongly in favor of the active group. Please rephrase it accordingly to reflect that the difference cannot be deemed as significant or that it is inconclusive.
---

VERSION 2 – AUTHOR RESPONSE

We have now amended the text on page 20 in response to the suggestion by reviewer 3 and we have removed the key and improved the labelling of Figure 2 in response to the comments from reviewer 1.

We hope that our amendments to the manuscript are satisfactory.